# Decarbonization of Transport and Oral Health

Morenike Oluwatoyin Folayan [1,2,3,4,*] and Maha El Tantawi [3,4,5]

1    Department of Child Dental Health, Obafemi Awolowo University, Ile-Ife 220282, Nigeria
2    Population Studies, Nigerian Institute of Medical Research, Lagos 101212, Nigeria
3    Early Childhood Caries Advocacy Group, University of Manitoba, Winnipeg, MB R3T 2N2, Canada;
     maha_tantawy@hotmail.com
4    Africa Oral Health Network, Alexandria University, Alexandria 21527, Egypt
5    Department of Pediatric Dentistry and Dental Public Health, Faculty of Dentistry, Alexandria University,
     Alexandria 21527, Egypt
*    Correspondence: toyinukpong@yahoo.co.uk

**Abstract:** The decarbonization of transport is a global initiative aimed at reducing greenhouse gas emissions and addressing the risks of global warming. This article explores the potential connections between the decarbonization of transport and oral health, highlighting the need for further research in this area. Emissions from vehicle exhausts, such as carbon dioxide, methane, and nitrous oxide, may have a modest impact on the risk of early childhood caries and other oral health diseases like periodontal diseases, oral cancer, and dental caries. Active transportation, which promotes regular exercise, has beneficial effects on overall health, including stimulating salivary protein production and reducing the risk of diabetes and cardiovascular diseases, both of which are linked to poor oral health. Transitioning to electric vehicles can also reduce noise pollution, positively impacting mental well-being, which is associated with improved oral hygiene practices. Furthermore, the development of sustainable infrastructure, including efficient public transportation systems, can enhance access to dental services. Further research is needed to establish stronger evidence for these connections and to explore how the global decarbonization of transport agenda can incorporate oral health considerations.

**Keywords:** climate change; global warming; greenhouse gas emissions; early childhood caries; regular exercise; salivary protein production; access to dental services





## 1. Introduction

The urgent need to address climate change and secure a sustainable future for our planet has driven the global push for decarbonization. One significant aspect of this effort is the decarbonization of transport, which involves reducing the carbon-intensive emissions produced by various modes of transportation that have detrimental effects on the environment and human well-being [1]. Transport emissions currently contribute about a quarter of the world's overall energy-related greenhouse gas emissions [2], with road transport alone accounting for 75% of total greenhouse gas emissions [3]. Given the rapid urbanization and increased motorization in developing nations, these emissions could rise by as much as 60% by the year 2050 [4].

However, the implications of decarbonizing transport go beyond its environmental benefits, as it has the potential to impact oral health. A shift to cleaner alternative energy can significantly reduce the risk for poor respiratory health, mouth breathing, dry mouth, and oral health problems like an increased risk of dental caries and periodontal diseases [5]. In addition, particulate matter that comes from vehicle emission is eliminated. Reduced exposure to such particulate matter due to cleaner transportation options can contribute to better oral health [6–9]. Furthermore, decarbonization efforts often encourage the adoption of active transportation modes like walking and cycling. These activities promote

activity and reduce sedentary lifestyles. A healthier lifestyle can positively impact oral health by reducing the risk of conditions like obesity and diabetes, which are linked to poor oral health outcomes [10,11]. Physical fitness and can lead to healthier lifestyle choices such as consumption of diet rich in fruits and vegetables [12], which supports oral health [13]. Finally, decarbonization efforts are accompanied by improved public transportation systems, which can enhance dental service utilization, thereby ensuring that dental problems are addressed in a timely manner [14]. Recognizing and understanding these multifaceted relationships can lead to more holistic approaches to public health initiatives that maximize this interconnectedness. We highlight the possible links between decarbonization of transport and oral health and the implications for future research on oral health.

### 1.1. Emissions from Vehicles and Oral Health

Emissions from vehicle exhaust, including carbon dioxide, methane, nitrous oxide, and particulate matter, have been found to have an influence on oral health [15]. Preliminary evidence suggests that the emission of greenhouse gases like carbon dioxide, methane, and nitrous oxide may have a modest effect on the risk of early childhood caries, with nitrous oxide specifically linked to its prevalence [16]. Fine and ultra-fine particulate matter released during combustion processes can serve as carriers for polycyclic aromatic hydrocarbons and toxic metals [17]. Polycyclic aromatic hydrocarbons are known to induce Reactive Oxygen Species in the body [18], leading to oxidative stress, cell membrane damage, lipid peroxidation, and subsequent cell dysfunction [19], increasing the risk of oral cancer [20]. Additionally, toxic metals present in oral fluid have been associated with oral diseases [21]. Prolonged exposure to high levels of transport emissions can heighten the risk of oral diseases, including periodontal diseases, oral cancer, and dental caries.

### 1.2. Active Transportation and Oral Health

Decarbonizing transport involves the promotion of active modes of transportation, such as walking and cycling, which not only contribute to the reduction of carbon emissions but also offer direct benefits for oral health [22]. Active transportation plays a crucial role in maintaining overall health. Regular exercise associated with active transportation activates the sympathetic nervous system and stimulates saliva production while regulating the secretion of various proteins from salivary glands [23–26] such as an increase in the concentration of S-type cystatins and cystatin C [27]. These cystatins in saliva help inhibit bacterial adhesion within the oral cavity by obstructing the binding of bacteria to buccal epithelial cells and inhibiting the growth of periodontopathic organisms without suppressing protease activity [28,29].

Additionally, after short and intense exercise, there is an immediate rise in the concentration or secretion rate of defense proteins in saliva, including α-amylase, lysozyme, lactoferrin, the cathelicidin LL-37, and α-defensins [30–32]. These proteins are essential for neutralizing acids produced by cariogenic bacteria, thereby potentially reducing the risk of dental caries [33,34]. Physical activity also reduces inflammatory biomarkers which, in turn, may improve periodontal health [35]. The concentration of these proteins in saliva changes over time following exercise, remaining elevated for up to 15 min after physical activity [27]. Moreover, increased physical activity associated with active transportation provides systemic health benefits, thereby reducing the risk of systemic conditions that can impact oral health, such as diabetes and cardiovascular diseases [36–38].

By promoting active modes of transportation, decarbonizing transport not only contributes to a healthier environment but also improves oral health through increased physical activity and the secretion of beneficial proteins in saliva. Of importance is the potential for endeavors aimed at promoting the decarbonization of transportation to yield improved oral hygiene and a reduced susceptibility to periodontal diseases and dental caries. Furthermore, the lasting advantages for oral health that stem from the cumulative impact of regular physical activity over an extended period are noteworthy. Through the integration

of heightened physical activity into patients' clinical oral healthcare strategies, healthcare professionals can effectively tap into the multifaceted advantages of physical exercise to enhance the outcomes of oral health. This, in turn, enables oral health clinicians to actively endorse and indirectly advocate for the decarbonization of transportation. This approach aligns harmoniously with a comprehensive approach to health that acknowledges the interconnected nature of diverse bodily systems. It underscores the notion that actions taken to enhance a specific facet of health, such as cardiovascular fitness, also exert positive influences on oral health and overall well-being.

### 1.3. Electric Vehicle and Oral Health

Furthermore, the transition towards electric vehicles embodies a significant advancement in the decarbonization of transport [39,40]. Unlike traditional internal combustion engines, electric vehicles produce zero tailpipe emissions, eliminating the harmful pollutants typically associated with air pollution [41]. By reducing exposure to these pollutants, the risk of oral diseases may be mitigated. Furthermore, the adoption of electric vehicles can indirectly contribute to improved oral health by reducing noise pollution [42]. This reduction in noise pollution can have a positive impact on mental well-being which is associated with better oral hygiene practices and overall oral health [43].

Previous research has provided evidence on the correlation between noise pollution and negative mental health outcomes, including depression, anxiety, and other mental disorders [44]. Chronic insomnia, which is associated with noise pollution, has also been identified as a risk factor for these mental health issues [45,46]. Furthermore, noise pollution has been linked to an increased risk of cardiometabolic diseases, including obesity [47,48]. These detrimental effects of noise pollution can have implications for oral health, contributing to an increased risk of oral health problems [49–53]. Consequently, the reduction in noise pollution achieved using electric vehicles can have a cascading effect, positively impacting mental well-being and indirectly leading to improved oral health outcomes.

### 1.4. Transportation Infrastructure Development and Oral Health

The decarbonization of transport necessitates the development of sustainable infrastructure, such as the establishment of bike lanes, pedestrian-friendly pathways, and efficient public transportation systems. This type of infrastructure not only promotes active modes of transportation but also facilitates access to dental care facilities. Insufficient transportation options have been identified as a significant barrier to accessing dental services, even for individuals who may not face significant financial barriers to care [54,55]. Concerns about transportation costs increases the likelihood of decreased dental visits [54]. By providing efficient public transportation systems, accessibility to timely and appropriate oral healthcare services can be improved, addressing an important aspect of oral health disparities.

## 2. Discussion

The adoption of low-carbon transport alternatives can lead to a reduction in the emission of harmful pollutants, resulting in improved oral health outcomes. Emphasizing active transportation as a means of addressing pollution, climate change, and overall health, including oral health, is crucial. Prior to the present study, the potential link between active transportation, climate change, and oral health has not been explored to the authors' knowledge.

The decarbonization of transport can contribute to the Sustainable Development Goal (SDG) 3 by promoting cost-effective methods of transportation, which is essential for reducing inequality as highlighted SDG 10. Additionally, investments in walking and cycling infrastructure serve as effective strategies for poverty reduction as highlighted by SDG 1. Furthermore, decarbonization significantly contributes to the reduction of greenhouse gas emissions, thus improving the climate and environment (SDG 13). These efforts also align with SDG 9, which focuses on the development of sustainable transportation infrastructure, and SDG 11, particularly its target of providing safe, affordable, accessible, and sustainable

transport systems for all, improving road safety, and expanding public transport options. By addressing these sustainable development goals, the decarbonization of transport can positively impact oral health outcomes while simultaneously benefiting various aspects of society and the environment.

Oral health is influenced by myriad factors, including oral hygiene practices [56], diet [57–59], socioeconomic status [60,61], and environmental exposures [62]. Environmental factors, such as air and noise pollution, negatively impact oral health outcomes [63–65]. However, the direct association between decarbonization of transportation and oral health is an emerging field of research, and more studies are needed to provide hard evidence of these links. These studies should be country and context specific taking cognizance of the different income levels. Disparities in the prevalence and access to oral health care differ by country income levels [66,67]. This research agenda responds to the 2021 World Health Assembly resolution calling for the incorporation of oral health within the non-communicable diseases' agenda and for a public health approach to address oral health problems [68], along with the 2023–2030 Global Strategy on Oral Health [69]. In addition, conducting studies on the link between transport decarbonization and oral health will ensure that oral health advocates are included in the discussions on the multi-sectoral solutions to population health and wellbeing and moves away from decades long narrow focus on teeth and oral cavity. Siloed endeavors to achieve the SDGs results from the failure to acknowledge the interconnectedness of the goals and the importance of their combined achievement for the wellbeing and prosperity of populations. Likewise, excluding critical components of health, like oral health, from the plans and goals for the decarbonization of transport as a strategic approach for healthy lives ignores the multiple connections between oral and general health.

The findings of this study suggest that the planning of decarbonization strategies in the transport sector should be carried out in conjunction with oral health experts. The December 2019 Conference on Health and Active Transportation brought together experts from the fields of transportation and health to discuss and shape the future of active transportation. The conference focused on Innovative Practices, Building Strategic Institutional Relationships, and Identifying Research Needs and Opportunities to pave the way for sustainable transportation solutions [22]. However, it is noteworthy that the conference outcomes and the derived conceptual model, which encompassed economic and systems analysis, evaluation of emerging technologies and policies, efforts to address inclusivity, disparities, and equity, as well as messaging and communication strategies, did not explicitly include an oral health perspective despite the strong relationship between oral health and overall health [70,71]. By integrating oral health into the discussions and planning processes surrounding decarbonization efforts in the transport sector, policymakers and stakeholders can ensure that oral health is not overlooked in the pursuit of sustainable and equitable transportation solutions.

Collaborative efforts between oral health experts, transportation planners, and policymakers can help identify specific research gaps and explore innovative strategies to promote oral health within the context of active transportation and sustainable transport infrastructure. Addressing oral health requires a multidimensional approach that integrates sustainable transportation practices and oral health promotion strategies. Collaboration among policymakers, urban planners, healthcare professionals, and community stakeholders. Such approaches may include incorporating oral health considerations into urban planning, promoting active transportation through public health initiatives, and integrating oral health education within sustainable transportation campaigns with the aim of improving access to oral health care. The recognition of this interconnectedness is vital for developing comprehensive and inclusive approaches that prioritize the oral and general well-being of individuals and communities as we strive for a sustainable and decarbonized future.

The bridges to link efforts to achieve sustainable and decarbonized transportation with oral health promotion activities should not exclude the peculiar needs of low- and

middle-income countries where these needs are acute. Sadly, most of the investment and research toward sustainable transport is currently directed toward high-income markets [72]. Briceno-Garmendia et al. [73] made the economic case for expanding electric vehicles to resource-limited settings and showed the significant opportunities for scaling up transport electrification using electric buses and two/three-wheelers. Such events may have a rippling effect on reducing oral health problems, which is currently skewed to resource-limited settings [74] Inclusive agenda are needed to build on previous experiences and avoid wasting of resources and time with detrimental impact on the wellbeing and prosperity of populations.

## 3. Conclusions

The decarbonization of transport is crucial not only for mitigating climate change but also for improving public health, including oral health. By transitioning to cleaner modes of transportation, such as active transportation and electric vehicles, we can significantly reduce air pollution, thereby decreasing the risk of oral diseases. Furthermore, the promotion of physical activity through sustainable transport infrastructure contributes to overall oral health and well-being. Therefore, the global decarbonization of transport agenda needs to actively include oral health considerations in its formulation and implementation.

**Author Contributions:** M.O.F. conceptualized the review and wrote the original draft preparation. M.E.T. reviewed the manuscript and edited the paper. All authors have read and agreed to the published version of the manuscript.

**Funding:** This research received no external funding.

**Institutional Review Board Statement:** Not applicable.

**Informed Consent Statement:** Not applicable.

**Data Availability Statement:** Not applicable.

**Conflicts of Interest:** The authors declare no conflict of interest.

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
