# Peer review of "Decarbonization of Transport and Oral Health"

_2673-8430, doi:10.3390/biomed3030032_

Round 1

Reviewer 1 Report

Dear authors,

congratulations on your manuscript on a very important topic.

I have a few suggestions- please relate the topic more in your introduction with oral health, remove the repetitive explanations from the discussion,   and explain the Active Transportation and oral health and Electric vehicle and oral health- parts in a more clinical aspect.

Make a flow chart diagram explaining the decarbonization and oral health.

english is fine.

Author Response

Thanks for the constructive feedback. Please find below a point-by-point response to the issues raised. The comments have helped strengthen the manuscript.

Comment 1: please relate the topic more in your introduction with oral health

We wrote in the introduction: A shift to cleaner alternative energy can significantly reduce the risk for poor respiratory health, mouth breathing, dry mouth, and oral health problems like an increased risk of dental caries and periodontal diseases [5]. In addition, particulate matter that comes from vehicle emission is eliminated. Reduced exposure to such particulate matter due to cleaner transportation options can contribute to better oral health [6-9]. Furthermore, decarbonization efforts often encourage the adoption of active transportation modes like walking and cycling. These activities promote activity and reduce sedentary lifestyles. A healthier lifestyle can positively impact oral health by reducing the risk of conditions like obesity and diabetes, which are linked to poor oral health outcomes [10, 11]. Physical fitness and can lead to healthier lifestyle choices such as consumption of diet rich in fruits and vegetables [12], which supports oral health [13]. Finally, decarbonization efforts are accompanied by improved public transportation systems, which can enhance dental service utilization, thereby ensuring that dental problems are addressed in a timely manner [14]. Recognizing and understanding these multifaceted relationships can lead to more holistic approaches to public health initiatives that maximized this interconnectedness. 

Comment 2: remove the repetitive explanations from the discussion

thanks for identifying this. We have removed the repetition and made the manuscript shorter. 

Comment 3: Explain the Active Transportation and oral health and Electric vehicle and oral health- parts in a more clinical aspect.

For the section on active transportation and oral health, we wrote: By promoting active modes of transportation, decarbonizing transport not only contributes to a healthier environment but also improves oral health through increased physical activity and the secretion of beneficial proteins in saliva. Of importance is the potential for endeavors aimed at promoting the decarbonization of transportation to yield improved oral hygiene and a reduced susceptibility to periodontal diseases and dental caries. Furthermore, the lasting advantages for oral health that stem from the cumulative impact of regular physical activity over an extended period are noteworthy. Through the integration of heightened physical activity into patients' clinical oral healthcare strategies, healthcare professionals can effectively tap into the multifaceted advantages of physical exercise to enhance the outcomes of oral health. This, in turn, enables oral health clinicians to actively endorse and indirectly advocate for the decarbonization of transportation. This approach aligns harmoniously with a comprehensive approach to health that acknowledges the interconnected nature of diverse bodily systems. It underscores the notion that actions taken to enhance a specific facet of health, such as cardiovascular fitness, also exert positive influences on oral health and overall well-being.

Make a flow chart diagram explaining the decarbonization and oral health.

Thanks for this suggestion. We are unable to make a flowchart of the literature as this was not a systematic review. 

Reviewer 2 Report

The scholarly document entitled "Decarbonization of Transportation and its Implications for Oral Health" offers a compelling and insightful exploration of the intersection between oral health care and the process of transport decarbonization. The manuscript exhibits a commendable scholarly composition, replete with pertinent citations, enhancing its credibility and informative capacity.

Author Response

Thanks for the positive comments

We are grateful
